# High Rates of Liver Cirrhosis and Hepatocellular Carcinoma in Chronic Hepatitis B Patients with Metabolic and Cardiovascular Comorbidities

**DOI:** 10.3390/microorganisms9050968

**Published:** 2021-04-30

**Authors:** Jan-Hendrik Bockmann, Matin Kohsar, John M. Murray, Vanessa Hamed, Maura Dandri, Stefan Lüth, Ansgar W. Lohse, Julian Schulze-zur-Wiesch

**Affiliations:** 1Department of Internal Medicine, University Medical Hospital Hamburg-Eppendorf, Martinistr. 52, 20246 Hamburg, Germany; Matinkohsar@outlook.de (M.K.); vanessa.hamed@immanuelalbertinen.de (V.H.); m.dandri-petersen@uke.de (M.D.); a.lohse@uke.de (A.W.L.); j.schulze-zur-wiesch@uke.de (J.S.-z.-W.); 2German Center for Infection Research (DZIF), Hamburg-Lübeck-Borstel Site, Martinistr. 52, 20246 Hamburg, Germany; 3School of Mathematics and Statistics, UNSW Sydney, Sydney 2052, Australia; j.murray@unsw.edu.au; 4University Medical Center Brandenburg, Center of Internal Medicine II, Brandenburg Medical School Theodor Fontane, Hochstr. 29, 14770 Brandenburg an der Havel, Germany; s.lueth@klinikum-brandenburg.de

**Keywords:** HBV, metabolic, NAFLD, HCC, cirrhosis

## Abstract

Background: The prevalence of metabolic and cardiovascular diseases is rising worldwide. However, little is known about the impact of such disorders on hepatic disease progression in chronic hepatitis B (CHB) during the era of potent nucleo(s)tide analogues (NAs). Methods: We retrospectively analyzed a single-center cohort of 602 CHB patients, comparing the frequency of liver cirrhosis at baseline and incidences of liver-related events during follow-up (hepatocellular carcinoma, liver transplantation and liver-related death) between CHB patients with a history of diabetes, obesity, hypertension or coronary heart disease (CHD). Results: Rates of cirrhosis at baseline and liver-related events during follow-up (median follow-up time: 2.51 years; NA-treated: 37%) were substantially higher in CHB patients with diabetes (11/23; 3/23), obesity (6/13; 2/13), CHD (7/11; 2/11) or hypertension (15/43; 4/43) compared to CHB patients without the indicated comorbidities (26/509; 6/509). Multivariate analysis identified diabetes as the most significant predictor for cirrhosis (*p* = 0.0105), while comorbidities did not correlate with liver-related events in pre-existing cirrhosis. Conclusion: The combination of metabolic diseases and CHB is associated with substantially increased rates of liver cirrhosis and secondary liver-related events compared to CHB alone, indicating that hepatitis B patients with metabolic comorbidities warrant particular attention in disease surveillance and evaluation of treatment indication.

## 1. Introduction

The prevalence of metabolic syndrome and the associated non-alcoholic fatty liver disease (NAFLD) is increasing worldwide and is becoming the most common cause of chronic liver disease in Western Europe and the US. In particular non-alcoholic steatohepatitis ( NASH) leads to increased all-cause mortality that is mainly related to cardiovascular disease and malignancy, including hepatocellular carcinoma (HCC) [1,2]. Likewise, chronic hepatitis B virus infection is a major health problem, with an estimated prevalence of HBsAg carriers of 3.61% worldwide. Approximately one-third of liver cirrhosis cases and half of HCCs can be attributed to hepatitis B virus infection [3,4]. While hepatitis C virus infection is considered to induce hepatic steatosis by elevating VLDL levels and insulin resistance [5,6,7], a promoting effect of CHB on liver steatosis or insulin resistance could not be demonstrated. Nevertheless, a low prevalence of NAFLD in CHB populations compared to healthy individuals has been described [8,9,10,11]. Furthermore, no negative effects of metabolic diseases on treatment response after interferon or NA treatment could be noticed in CHB patients [12,13]. NAs prevent the progression of liver disease towards cirrhosis by inducing effective suppression of viral replication [14,15] but have little effect on HBsAg levels. Moreover, single cases of HCC development in NA treated patients have been reported [16]. The mechanisms leading to HCC development in these patients are not fully understood. Amidst a rising prevalence of metabolic syndrome, it becomes key to assess whether metabolic diseases may play a role in cirrhosis and HCC development in CHB populations harboring low HBV DNA levels (as inactive carriers or treated patients). Several studies, mainly conducted in Asian countries, have analyzed the impact of metabolic factors on hepatitis B disease progression [17,18,19,20,21,22,23]. However, little data are yet available on extrahepatic metabolic diseases associated with unfavorable hepatitis B disease outcomes in patients treated according to current European guidelines [24]. Therefore, this study aimed to investigate whether a comprehensive set of metabolic comorbidities and other clinical parameters correlate with the occurrence of cirrhosis and severe liver-related events (HCC, liver-related death, liver transplantation) in a large German single-center cohort of CHB patients.

## 2. Materials and Methods

In this study, we screened 651 hepatitis B patients who attended the University Medical Center Hamburg-Eppendorf between January 2008 and June 2017 for the first time. The inclusion criterion was positive HBsAg status for at least 6 months. A total of 13 patients presenting a history of HCC or liver transplantation before baseline were excluded from the study. 36 CHB patients with positive serum HDV or HCV RNA determined by PCR at any time point, or presenting a history of HIV coinfection, were also excluded. Six hundred two patients were included in the subsequent analysis (Figure 1). Virological (serum HBV DNA, HBsAg, HBeAg status) and biochemical parameters (alanine aminotransferase (ALT), aspartate aminotransferase (AST), gamma-glutamyltransferase (GGT), alkaline phosphatase (AP), INR, bilirubin, albumin, platelet count, creatinine) were retrospectively analyzed by detailed chart review. Diagnosis of liver cirrhosis was based on liver histology (F4 according to the Desmet score), if available, or by transient elastography (≥13 kilopascals) [25]. Hepatocellular carcinoma was diagnosed according to the respective national guideline either by biopsy or cross-sectional imaging with computed tomography and magnetic resonance imaging scan [26,27]. Obesity was determined by a body mass index > 30 kg/m^2,^ and non-alcoholic fatty liver disease was diagnosed by ultrasound.

Statistical analysis was performed with SPSS and Matlab R2017a. Figures were created by using the GraphPad Prism software (Prism 9.0.0, released on October 2020). Parameters of different patient groups were compared by the nonparametric Mann–Whitney *U*-test and the chi-squared test. The association between parameters at baseline and disease-related clinical events was analyzed by univariate and multivariate logistic regression. A value of *p* < 0.05 was considered statistically significant. The study was approved at the local ethics board (Ethik-Kommission der Ärztekammer Hamburg, WF-035/17, March 2017) and conformed to German law and the principles espoused in the Declaration of Helsinki.

## 3. Results

### 3.1. High Rates of Liver Cirrhosis in Patient Subgroups with Metabolic and Cardiovascular Comorbidities

Here, we analyzed the detailed clinical outcome of 602 CHB patients showing positive HBsAg levels for at least six months and attending our university viral hepatitis clinic between January 2008 and June 2017. 50 (8%) patients presented with cirrhosis, 225 (37%) were NA treated, 30 (5%) were IFN-treated and 198 presented as untreated HBeAg -negative CHB (phase III according to EASL guidelines 2017). Retrospective chart review revealed the following frequencies of patients with a history of metabolic diseases at baseline: 23 patients with diabetes type II, 13 with obesity, 11 with coronary heart disease, 43 with hypertension and 38 with non-alcoholic fatty liver disease. The baseline characteristics of the overall cohort and metabolic subgroups are summarized in Table 1. Hepatitis B viremia and HBsAg levels did not differ significantly between these groups. The median (95% CI) observation period of 2.51 (2.28–2.68) years for the overall cohort was quite comparable to those of subgroups NAFLD (2.26, 1.96–3.01), obesity (2.01, 0.46–3.65), coronary heart disease (2.55, 0.41–4.96), diabetes (2.09, 1.55–3.09) and hypertension (2.22, 1.63–2.98). Rates of liver cirrhosis in metabolic subgroups diabetes (*p* < 0.0001), obesity (*p* < 0.0001), coronary heart disease (*p* < 0.0001) and hypertension (*p* < 0.0001) were significantly higher than the overall cohort, but not different between the overall cohort and the NAFLD subgroup (*p* = 0.5772). Since age and male sex differed significantly between subgroups and the overall cohort (Table 1), we analyzed the correlation between baseline characteristics and cirrhosis by logistic regression. Univariate regression analysis showed that sex, age, HBeAg status, NA therapy, IFN therapy, diabetes, obesity, CHD and hypertension positively correlated with the occurrence of cirrhosis (Table 2). Multivariate logistic regression was performed, including 9 significant variables. A stepwise multivariate regression analysis presented diabetes as a significant and independent predictor for cirrhosis.

### 3.2. Clinical Outcome of CHB Patients with Different Metabolic Comorbidities

Liver-related events (hepatocellular carcinoma, liver-related death, liver transplantation) occurred in 14/602 (2.33%) patients of the overall cohort (13/602, 7/602 and 6/602, respectively) during follow-up. Interestingly, liver-related events occurred more often in subgroups with diabetes (3/23, 13.04%, *p* = 0.0046), obesity (2/13, 15.38% *p* = 0.0054), hypertension (4/43, 9.30%, *p* = 0.0085) and coronary heart disease (2/11, 18.18%, *p* = 0.0011) than in the overall cohort (Appendix A). By contrast, no liver-related events occurred in the NAFLD group. In addition, normalization of total liver-related events to the observation period resulted in higher event rates/1000 person-years within metabolic subgroups obesity (69 events/1000 person-years), coronary heart disease (64 events/1000 person-years), diabetes (56 events/1000 person-years) and hypertension (36 events/1000 person-years) compared to the overall cohort (8.5 events/1000 person-years). Taken all patients with cirrhosis together, those with (7/24) and without (6/26) metabolic diseases (*p* = 0.6238) did also not differ concerning the occurrence of liver-related events (7/24 vs. 6/26, *p* = 0.6238), while among non-cirrhotic patients, 1 case of HCC occurred in the total metabolic group (1/69), but none in those patients, who did not have metabolic comorbidities (0/483, *p* = 0.0081) (Figure 1). Total liver related-events were mainly determined by the occurrence of HCCs, which were more frequent in subgroups with diabetes (3/23, 13.04%, *p* = 0.0012) and obesity (2/13, 15.38% *p* = 0.0036) compared to the overall cohort, while coronary heart disease (1/11, 9.09%, *p* = 0.1273) and hypertension (3/43, 6.97%, *p* = 0.0549) subgroups did not significantly differ from the overall cohort concerning HCC rates. 13/14 patients with liver-related events presented with liver cirrhosis at baseline, so that most of the liver-related events occurred secondary to liver cirrhosis. Hence, we investigated whether baseline characteristics in the subcohort of CHB patients with cirrhosis (Table 1) correlated with the occurrence of liver-related events, thus providing predictive factors for hepatitis B disease outcome in cirrhosis. Univariate logistic regression analysis revealed that baseline values of age (*p* = 0.0345) and platelet count (*p* = 0.0363) correlated significantly with the occurrence of liver-related events during follow-up (Appendix A). However, diabetes, obesity, CHD, hypertension and NAFLD did not correlate with liver-related events as well as sex, HBeAg status, ALT, GGT, albumin or MELD score. Multivariate stepwise regression model identified low platelets (*p* = 0.0403) as the most significant independent variable for total liver-related events and age (*p* = 0.0273) for HCC events in cirrhotic CHB patients (Appendix A).

## 4. Discussion

The current analysis of a large single-center cohort describes a relatively low frequency of different metabolic and cardiovascular diseases in CHB patients in line with previous studies describing the low frequency of fatty liver disease in CHB [8]. During long-term follow-up, the clinical outcome of these patients was then substratified according to the presence or absence of metabolic or cardiovascular comorbidities in long-term follow-up. Indeed, subgroups with diabetes, obesity, CHD and hypertension were more likely suffering from liver cirrhosis, while diabetes was the only independent variable that correlated with cirrhosis development in multivariate analysis. In addition, liver-related events occurred significantly more often in all 4 subgroups. Since most (13/14) of these events occurred in cirrhotic patients and since liver event rates did not correlate with the indicated subgroups in patients with pre-existing cirrhosis, our data would indicate that such high event rates in CHB patients with metabolic comorbidities were determined by progression to liver cirrhosis per se and did not increase the risk of HCC, death or liver transplantation in pre-existing cirrhosis. Accordingly, univariate and multivariate analysis of cirrhotic CHB patients identified age and thrombocytopenia—but not comorbidities-as the most significant predictors for liver-related events, reflecting 2 of the main PAGE-B score variables (age, low platelets and male sex), while male sex was not associated with the occurrence of liver-related events in this study.

In particular, CHB patients with metabolic comorbidities developed significantly more HCCs than CHB patients without comorbidities (7.5% vs. 1.2%). It is important to note that the overall HCC risk determined in meta-analyses of large Asian and US patient NAFLD/NASH cohorts, including patients with metabolic and cardiovascular comorbidities, ranged between 1.8/1000 person-years [28] and 5.29/1000 person-years [29] with advanced fibrosis as the main risk factor. In this study, the HCC rate in total hepatitis B patients treated according to the current European guidelines was higher (9.92 events/1000 person-years) than those of indicated NAFLD/NASH cohorts but comparable to previously described CHB cohorts [30,31]. Notably, CHB patients with metabolic comorbidities in this study showed remarkably higher HCC rates (36–69/1000 person-years) than both the overall hepatitis B cohort of this study and NASH cohorts with advanced liver fibrosis in previous studies [28]. Taken together, our data indicate that rates of HCCs in hepatitis B patients with metabolic comorbidities are higher than in patients suffering from either hepatitis B or metabolic diseases alone, whereby elevated HCC risks can be attributed to the higher cirrhosis rates in these patients.

In this context, it was quite astonishing that the frequency of liver cirrhosis and secondary liver-related events in the NAFLD subgroup was not higher than in the overall cohort. The role of hepatic steatosis in chronic hepatitis B remains controversial [32,33]. The interesting question of whether the subtype of non-alcoholic steatohepatitis is a more relevant predictor for the outcome in HBV infection cannot be answered by this study, as liver biopsies were not available for most of the NAFLD patients. Furthermore, alcohol consumption was not systematically assessed by this study, so that formally alcoholic fatty liver disease cannot be ruled out. However, alcohol consumption was assessed as generally to be minimal in this cohort according to the providers and patient charts. Admittedly, mild ALT levels in the NAFLD subgroup indicate that most of this group did not suffer from higher degrees of steatohepatitis. Indeed, some previous studies have described steatohepatitis as a stronger predictor for poor clinical outcome in CHB than mild steatosis alone [32,34].

In summary, the data of this study demonstrate higher rates of liver cirrhosis in patients with CHB and metabolic/cardiovascular comorbidities—in particular diabetes—compared to CHB without comorbidity. However, high rates of liver-related events in these patients are related to liver cirrhosis per se—age and low platelets may thereby serve as driving risk factors. Consequently, hepatitis B patients with metabolic comorbidities warrant particular attention in disease surveillance and evaluation of treatment indications. An appealing question would be if extending the indication for antiviral therapy in CHB patients with metabolic diseases would decrease the risk of progressive liver disease, particularly if the therapy of comorbidities is ineffective. Furthermore, metabolic comorbidities like diabetes may add a further dimension to improve HCC predictive scores like PAGE-B. Further prospective studies and a deeper understanding of the underlying molecular and inflammatory mechanisms are needed to characterize the disease course and treatment success of CHB subgroups with such comorbidities.

## Figures and Tables

**Figure 1 microorganisms-09-00968-f001:**
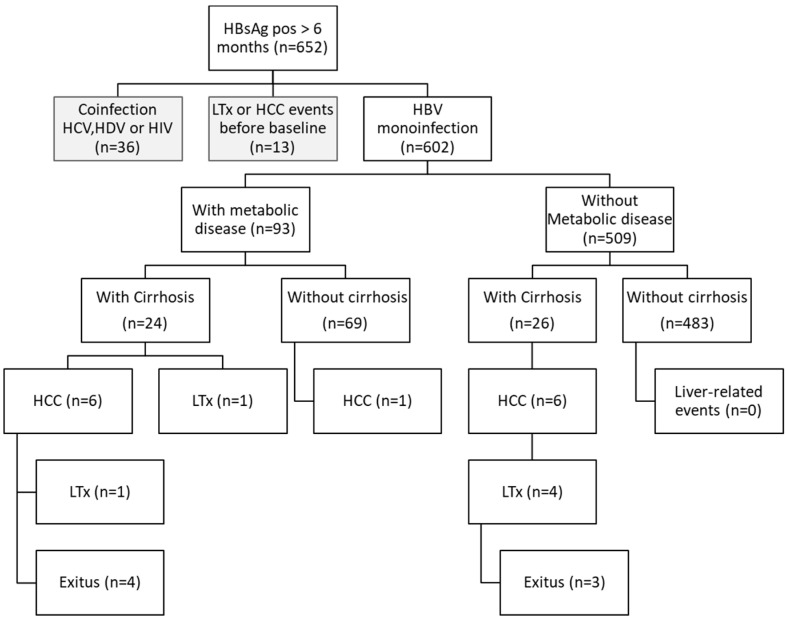
Patient outcome of chronic hepatitis B patients with and without metabolic diseases. Liver-related events included exitus letalis, hepatocellular carcinoma and liver transplantation. HBsAg = hepatitis B surface antigen, HCV = hepatitis C virus, HDV = hepatitis D virus, HIV = human immunodeficiency virus, HCC = hepatocellular carcinoma, LTx = liver transplantation.

**Table 1 microorganisms-09-00968-t001:** Baseline characteristics of total CHB patients and subgroups with diabetes, obesity, coronary heart disease (CHD), hypertension and non-alcoholic fatty liver disease (NAFLD). IQR = interquartile range, NA = nucleo(s)tide analogues, IFN = interferon, HBV = hepatitis B virus, HBsAg = hepatitis B surface antigen, HBeAg = hepatitis B early antigen, INR = international normalized ratio, ALT = alanine aminotransferase, GGT = gamma-glutamyltransferase.

	Total *n* = 602	With Cirrhosis *n* = 50	Without Cirrhosis *n* = 552	Diabetes *n* = 23	Obesity *n* = 13	CHD *n* = 11	Hypertension *n* = 43	NAFLD *n* = 38
Male sex	341 (56.64%)	38 (76%)	303 (54.89%)	18 (78.26%)	6 (46.15%)	11 (100%)	38 (88.37%)	27 (71.05%)
Median age, years (IQR)	42 (33.25–52)	58.5 (50–64)	40 (32–50)	57 (50.5–64)	48 (46–61)	62 (58–71.5)	58 (50–67)	46 (41–54.5)
Cirrhosis	50 (8.31%)	50 (100%)	0 (0%)	11 (47.83%)	6 (46.15%)	7 (63.64%)	15 (34.88%)	4 (10.53%)
NA treatment during follow-up	225 (37.38%)	42 (84%)	183 (33.15%)	14 (60.87%)	9 (69.23%)	11 (100%)	41 (95.35%)	15 (39.47%)
IFN treatment during follow-up	30 (4.98%)	8 (16%)	22 (3.99%)	2 (8.7%)	0 (0%)	1 (9.09%)	4 (9.30%)	1 (2.63%)
Median HBV DNA, IU/mL (IQR)	1.71 × 10^3^ (1.2 × 10^2^–5.17 × 10^4^)	6.03 × 10^3^ (1.2 × 10^1^–8.19 × 10^5^)	1.72 × 10^3^ (1.2 × 10^1^–5.17 × 10^4^)	5 × 10^2^ (1.25 × 10^1^–1.51 × 10^5^)	5.17 × 10^2^ (1.2 × 10^1^–1.72 × 10^5^)	2 × 10^2^ (1.2 × 10^1^–2.59 × 10^4^)	9.48 × 10^1^ (1.2 × 10^1^–4.31 × 10^4^)	1.44 × 10^3^ (1.2 × 10^1^–3.45 × 10^4^)
Median HBsAg, IU/mL (IQR)	5.05 × 10^3^ (5 × 10^2^–1.38 × 10^4^)	6.32 × 10^3^ (5 × 10^2^–1.21 × 10^4^)	4.66 × 10^3^ (5 × 10^2^–1.46 × 10^4^)	2.74 × 10^3^ (5 × 10^2^–1.72 × 10^4^)	6.4 × 10^3^ (5 × 10^2^–1.16 × 10^4^)	3.25 × 10^3^ (5 × 10^2^–8.35 × 10^3^)	2.59 × 10^3^ (5 × 10^2^–8.35 × 10^3^)	2.19 × 10^3^ (5 × 10^2^–1.02 × 10^4^)
HBeAg-positive	56 (9.3%)	12 (24%)	44 (7.97%)	7 (30.43%)	3 (23.08%)	3 (27.27%)	4 (9.30%)	3 (7.89%)
Median INR (IQR)	1.02 (0.97–1.09)	1.13 (1.05–1.27)	1.01 (0.97–1.07)	1.06 (1.01–1.86)	1.06 (1.01–1.08)	1.2 (1.07–1.25)	1.07 (0.99–1.14)	1.11 (0.97–1.05)
Median bilirubin, mg/dL (IQR)	0.5 (0.4–0.7)	0.7 (0.5–1.2)	0.5 (0.4–0.7)	0.5 (0.4–0.8)	0.5 (0.4–0.7)	1 (0.6–1.7)	0.6 (0.4–0.9)	0.5 (0.4–0.8)
Median platelets, MRTD/L (IQR)	217 (176.75–260)	136 (80–186)	221 (184–266)	179 (112–220.5)	202 (113–259)	113 (67.25–183.5)	198 (125.25–242.5)	217 (194–279.5)
Median ALT, U/L (IQR)	31 (22–53.5)	53 (33–94)	31 (21–50)	51 (27–52.5)	38 (32–59)	49 (30–79.75)	35 (23–51.25)	36 (27–53)
Median GGT, U/L (IQR)	27 (19–48)	84 (43–202)	26 (19–43)	66 (31.5–92.5)	72 (24–184)	79 (68.25–177.5)	58 (26–158)	49 (27–72)
Median albumin, g/L (IQR)	43 (40–46)	39 (35–41)	43 (40–46)	40.5 (39.75–43.25)	40 (40–42)	37 (29.75–42.5)	40 (38–44)	44 (40–46.75)

**Table 2 microorganisms-09-00968-t002:** Correlation between baseline parameters and occurrence of cirrhosis analyzed by univariate and multivariate logistic regression. CHD = coronary heart disease, NA = nucleo(s)tide analogues, IFN = interferon, HBeAg = hepatitis B early antigen.

	Univariate Analysis (*p*-Value)	Multivariate Analysis (*p*-Value)
Sex	0.0031	0.61528
Age	<0.001	<0.001
NA therapy	<0.001	0.0014
IFN therapy	<0.001	0.1885
HBeAg status	<0.001	0.0372
Diabetes	<0.001	0.0105
Obesity	0.0048	0.8070
CHD	<0.001	0.8698
Hypertension	<0.001	0.0801

## Data Availability

The datasets used and/or analyzed during the current study are available from the corresponding author upon reasonable request.

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
