# Peer review of "High Rates of Liver Cirrhosis and Hepatocellular Carcinoma in Chronic Hepatitis B Patients with Metabolic and Cardiovascular Comorbidities"

_microorganisms, 2021, doi:10.3390/microorganisms9050968_

Round 1

Reviewer 1 Report

The authors analyzed the frequency of liver cirrhosis and liver-related events in patients with chronic hepatitis B and diabetes/obesity/hypertension or coronary heart disease.

The manuscript is well organized, just some minor improvements need to be made, editing, typos, abbreviations:

  • better full word Hypertension in Table 1
  • verify the use of abbreviated words - yGT, NUC, and others
  • explanation of the abbreviations in the legend of the Tables 
  • In Figure 1, better to write clearly Without metabolic disease, Without cirrhosis, and to explain the abbreviations

Author Response

We thank reviewer 1 for her/his benevent reception of the study. See responses below.

1) better full word Hypertension in Table 1

We agree. Abbreviation “Hypert.” has been deleted and changed for “Hypertension” in the revised manuscript (page 3, Table 1).

2) verify the use of abbreviated words - yGT, NUC, and others

Abbreviation words have now been verified in the material section (page 2, line 71-72) and a list of abbreviations has been added (page 7, line 237-243) in the revised version of the manuscript.

3)  explanation of the abbreviations in the legend of the Tables 

Also see above. Please find the abbreviations in the legends of the revised Table 1 and 2 as wells as Supplementary Table 1.

4)  In Figure 1, better to write clearly Without metabolic disease, Without cirrhosis, and to explain the abbreviations.

Figure 1 has been changed accordingly and abbreviation explanations have been added to the legend.

Reviewer 2 Report

The manuscript by Bockmann and colleagues investigated the clinical impact of metabolic comorbidities in patients with chronic hepatitis B. Given the growing prevalence of metabolic syndrome and NAFLD especially in western countries, the topic assessed is relevant. However, I have important concerns with the methodological approach pursued by the authors.

Major comments:

1) The overall statistical analysis could have been done in a more linear an simple way. To investigate the association between metabolic comorbidities and the presence of "baseline" cirrhosis, patients should be divided in those with and without cirrhosis, and then authors should have performed a regression analysis (univariate and multivariate) to investigate the association between variables (of course including the different metabolic comorbidities) and the presence of cirrhosis. Accordingly, Table 1 should  report the characteristics of the overall cohort of patients and two more columns with the same characteristics according to the presence or not of cirrhosis.

2) As above, authors should use survival curve analysis (univariate test) followed by a cox regression analysis (multivariate) to evaluate the association between baseline characteristics of patients and the incidence of liver-related events. Therefore, in my opinion, there is no need for a propensity score matching.

3) Patients. Overall 602 patients were included in the study. 50 (8.31%) had cirrhosis. 225 (37%) were under NAs treatment. 30 (5%) were treated with IFN during FU. How many inactive carriers (HBeAg-negative chronic infection) were included in the analysis? It is important to provide the phases of HBV chronic infection (see EASL guidelines 2017) for the patients enrolled.

4) For such study, the low number of patients with cirrhosis at baseline and the low mean FU could represent an important limitation. Indeed, the overall numer of patients with liver events is pretty low (14 out of 602).  

Minor comments:

1) After amending the analysis of the study, please provide in the abstract the estimates of the associations and the corresponding p values.

2) Materials and methods. Line 73. Please add a reference for the cut-off of liver stiffness selected.

3) There is no mention on alcohol consumption. Please add.

4) Mean FU. Was the variable normally distributed? Which statistical test was used to assess normality? Maybe, reporting the years of FU as median and 95% CI or range would be more correct.

Author Response

We thank the reviewer for the useful suggestions. Please find our point-by-point responses below.

Major comments:

1) The overall statistical analysis could have been done in a more linear an simple way. To investigate the association between metabolic comorbidities and the presence of "baseline" cirrhosis, patients should be divided in those with and without cirrhosis, and then authors should have performed a regression analysis (univariate and multivariate) to investigate the association between variables (of course including the different metabolic comorbidities) and the presence of cirrhosis. Accordingly, Table 1 should  report the characteristics of the overall cohort of patients and two more columns with the same characteristics according to the presence or not of cirrhosis.

We see the reviewer´s point. We consulted with our expert Prof. John Murray (School of Mathematics and Statistics, UNSW Sydney, Australia). We recalculated the entire data set: A correlation between baseline characteristics (obesity, diabetes, coronary heart disease, hypertension, steatosis hepatis, sex, age, HBeAg status, treatment with nucleotid analogue, treatment with Interferon) and cirrhosis has now been performed for the total cohort by univariate and multivariate logistic regression.  Biochemical parameters (platelet count, albumin, INR, MELD score, bilirubin, ALT, AST, yGT) were not included into the analysis since they are directly linked to cirrhosis. Indeed, multivariate analysis highlights diabetes as an independent predictor for cirrhosis. The new results have been incorporated in the results section (page 3, line 107-113) of the revised manuscript and are presented in Table 2. Furthermore, 2 additional columns with baseline characteristics of CHB patients with and without cirrhosis have been added to table 1 as suggested.

2)  As above, authors should use survival curve analysis (univariate test) followed by a cox regression analysis (multivariate) to evaluate the association between baseline characteristics of patients and the incidence of liver-related events. Therefore, in my opinion, there is no need for a propensity score matching.

We thank reviewer 2 for this suggestion: Correlation of baseline characteristics with the occurrence of liver-related events has been reperformed by univariate as well as multivariate analysis in the subgroup of cirrhosis patients. The results are shown in Supplementary Table 1 of the revised manuscript. Since most of the events occurred in the cirrhotic subgroup, we think that multivariate analysis within the non-cirrhotic subgroup is not appropriate. The new data are reported in the results section (page 5, line 143-155). In line with data of the submitted manuscript, metabolic comorbidities do not correlate with hepatic events in pre-existing cirrhosis. The propensity score analysis has been removed as proposed by the reviewer.

3) Overall 602 patients were included in the study. 50 (8.31%) had cirrhosis. 225 (37%) were under NAs treatment. 30 (5%) were treated with IFN during FU. How many inactive carriers (HBeAg-negative chronic infection) were included in the analysis? It is important to provide the phases of HBV chronic infection (see EASL guidelines 2017) for the patients enrolled.

We defined the phases of chronic hepatitis B in untreated patients at baseline. 364 of these patients were HBeAg-negativ with 198 in phase III and 166 in phase IV according to the EASL guidelines 2017. The data are reported in the revised results section (page 3, line 93-95).

4) For such study, the low number of patients with cirrhosis at baseline and the low mean FU could represent an important limitation. Indeed, the overall number of patients with liver events is pretty low (14 out of 602).  

The reviewer raises an important issue. We agree, that the low event rate (14/602) may be based on a short observation period. Furthermore, low number of events in the subgroups may result in underestimated differences in event rates between metabolic subgroups. Thus, we avoided to overinterpret such differences and discussed the limitations especially for the NAFLD subgroup. However, it is important to note that differences in cirrhosis (26% vs 5%, p<0.0001) and liver event rates (9% vs 1%, p<0.0001) between CHB patients with and without metabolic diseases are quite significant in this analysis. Furthermore, diabetes was shown to be an independent predictor for cirrhosis  in comprehensive multivariate analyses.

Minor comments:

1) After amending the analysis of the study, please provide in the abstract the estimates of the associations and the corresponding p values.

The abstract has been edited (page 1, line 26-27).

2) Materials and methods. Line 73. Please add a reference for the cut-off of liver stiffness selected.

We have added a new reference in the material section (page 2, line 75).

3) There is no mention on alcohol consumption. Please add.

We now discuss this point in the revised manuscript (page 7, line 216-219).

4) Mean FU. Was the variable normally distributed? Which statistical test was used to assess normality? Maybe, reporting the years of FU as median and 95% CI or range would be more correct.

Please find the median FU time and 95% CI in the revised abstract (page 1, line 23) and in the revised results section (page 3, line 100-103). The analysis was performed by GraphPad Prism 9.

Round 2

Reviewer 2 Report

The authors improved the manuscript according to the comments raised. In my opinion, the manuscript is now suitable for publication in Microorganisms.